# Therapeutic Potential of *Bifidobacterium longum* subsp. *infantis* B8762 on Gut and Respiratory Health in Infant

**DOI:** 10.3390/ijms26031323

**Published:** 2025-02-04

**Authors:** Rocky Vester Richmond, Uma Mageswary, Adli Ali, Fahisham Taib, Thai Hau Koo, Azianey Yusof, Intan Juliana Abd Hamid, Feiyan Zhao, Nik Norashikin Nik Abd Rahman, Taufiq Hidayat Hasan, Heping Zhang, Min-Tze Liong

**Affiliations:** 1Department of Pediactric, Faculty of Medicine, Universiti Kebangsaan Malaysia, Kuala Lumpur 56000, Malaysia; rockyvr96@gmail.com; 2School of Industrial Technology, Universiti Sains Malaysia, Gelugor 11800, Malaysia; umamageswary1901@gmail.com; 3Pediatric & Palliative Care, Hospital Universiti Sains Malaysia, Kota Bharu 16150, Malaysia; fahisham@usm.my (F.T.); waynehau25@gmail.com (T.H.K.); 4Kepala Batas Health Clinic, Ministry of Health Malaysia, Putrajaya 13200, Malaysia; azianeyyusof@gmail.com; 5Advanced Medical & Dental Institute, Universiti Sains Malaysia, Gelugor 13200, Malaysia; intanj@usm.my; 6Key Laboratory of Dairy Biotechnology and Engineering, Ministry of Education, Inner Mongolia Agricultural University, Hohhot 010018, Chinahepingdd@vip.sina.com (H.Z.); 7Wakaf Che Yeh Health Clinic, Ministry of Health Malaysia, Kota Bharu 15100, Malaysia; nikariffudin01@gmail.com; 8IIUM Medical Centre, International Islamic University Malaysia, Kuantan 25200, Malaysia; dr_taufiq@iium.edu.my

**Keywords:** probiotic, *Bifidobacterium longum* subsp. *infantis* B8762, gut health, respiratory illness, immunomodulation, pediatrics

## Abstract

Respiratory tract and gastrointestinal infections in pediatric populations are major public health concerns. Addressing these challenges necessitates effective preventative and therapeutic strategies. This study assessed the efficacy of the probiotic *Bifidobacterium longum* subsp. *infantis* B8762 (0.5 × 10^10^ CFU) in reducing the duration and frequency of these infections in young children. In a randomized trial, 115 eligible children were assigned to either the probiotic (*n* = 57; 3.51 ± 0.48 months old) or placebo (*n* = 58; 2.78 ± 0.51 months old) group, with daily consumption for 4 weeks. The probiotic group demonstrated a lower duration of infections than the placebo group (*p* < 0.05). The probiotic group also showed fewer clinical visits due to respiratory and gastrointestinal problems as compared to the placebo group (*p* = 0.009 & *p* = 0.004, respectively). Oral swab samples revealed that the placebo group had higher levels of pro-inflammatory cytokine TNF-α after 4 weeks (*p* = 0.033), while the probiotic group demonstrated a balanced cytokine response, indicating modulation of the immune system. Genomic analysis showed that B8762 harbors various genes for the synthesis of proteins and vitamins crucial for the gut health of children. Both the clinical and genomic findings suggested that B8762 offered a therapeutic effect on gut and respiratory health in children, highlighting its potential in managing common pediatric infections.

## 1. Introduction

Respiratory tract infections and gastrointestinal illnesses such as diarrhea are more common in children than in adolescents and the adult population. In fact, data from the Ministry of Health Malaysia showed that the conditions accounted for 5.6% and 1.0% of deaths among children under five years old, respectively [1]. These illnesses often result in substantial morbidity, leading to increased healthcare utilization, parental absenteeism, and a considerable socioeconomic burden. Two Malaysian studies have highlighted concerning trends in the incidence of both diseases. One study recorded a 36.3% rise in respiratory syncytial virus (RSV) infections in the post-COVID-19 era compared to pre-pandemic levels, while another found that Malaysia’s constant humidity and rainfall contribute to seasonal peaks of rotavirus-related diarrhea, exacerbating gastrointestinal illness challenges [2,3]. Given the high incidence, recurrent nature, and persistent prevalence of both respiratory and gastrointestinal infection, there is an urgent need for effective prevention and treatment strategies to mitigate their impact on pediatric populations.

The gut microbiota, composed of trillions of microorganisms, plays a crucial role in immune regulation such as modulating the activity of immune cells such as T-cells and regulatory T-cells [4]. These immune cells can travel from the gut to the lungs, where they help control inflammation and protect against respiratory infections. Additionally, the airway mucosal surfaces are inhabited by lung microbiota, made up of beneficial commensal microorganisms that interact with the host immune system, particularly during respiratory viral infections [5]. Gambadauro et al. further reviewed, noting that lung microbiota composition affects the severity of viral infections and the immune response, with microbial metabolites influencing the activation of local immune cells during inflammation [5]. The regulation of immunomodulation and the maintenance of pulmonary homeostasis through healthy bacteria via gut–lung interaction is supported using beneficial probiotics. Probiotics are defined as “live microorganisms, which, when consumed in adequate amounts, confer a health effect on the host” [6]. They help promote a balanced microbiota, which enhances immune function and modulates the gut–lung axis (the bidirectional communication between the gut and lung microbiota), providing a prophylactic effect. In pediatric populations, the immunomodulatory properties of probiotics are particularly beneficial for therapeutic purposes. Probiotics bolster the immune system by stimulating antimicrobial substance production, activating immune cells, and promoting the secretion of anti-inflammatory cytokines [7]. These actions are essential in defending against pathogens, reducing the incidence and severity of respiratory and gastrointestinal illnesses in children, and maintaining overall health and immune homeostasis. These actions are critical in defending against pathogenic invasions, thereby reducing the incidence and severity of respiratory and gastrointestinal illnesses in children, hence maintaining the overall health and immune homeostasis.

Clinical studies have provided compelling evidence supporting the efficacy of probiotics in reducing the incidence and duration of respiratory illnesses and gastrointestinal symptoms in children. For instance, randomized controlled trials have demonstrated that children receiving probiotic supplementation experience fewer episodes of respiratory infections, reduced prescriptions of antibiotics, a shorter duration of illness, and reduced hospitalization periods compared to those receiving a placebo [8]. Similarly, Grandy et al. had observed a reduction in vomiting duration in intervention groups receiving probiotics during acute rotavirus diarrheal episodes [9]. The continued exploration of specific probiotic strains and their mechanisms of action is essential to fully elucidate the potential of probiotics in preventing respiratory and gastrointestinal illnesses in pediatric populations. Tailored probiotic therapies, based on individual strains and specific health needs, could further enhance the efficacy of these interventions [10].

*Bifidobacterium* spp., first isolated in 1899, is dominant in the gut microbiota of breastfed infants. Among the various subspecies, *B. infantis* has demonstrated superior beneficial potential compared to others such as *B. longum* subsp. *longum*, *B. breve*, *B. bifidum*, and *B. adolescentis* [11]. Specific strains of *B. infantis*, including M63, ATCC 15697, UCD272, EVC001, BB02, BT1, and R0033, have been clinically proven to reduce gut dysbiosis, lower pH, and potentially play a role in the development and maturation of the immune system [11]. While numerous strains of Bifidobacterium have been extensively studied, the efficacy of these strains remains highly strain-dependent, and robust clinical evidence is still needed to confirm their therapeutic potential. Recent interest in probiotics, particularly the use of strains like *B. longum* subsp. *infantis* B8762 [8,12] as adjunctive therapies, has further emphasized their potential to modulate the gut microbiota. However, the specific impact of *B. longum* subsp. *infantis* B8762 on respiratory illnesses in children remains unexplored. Furthermore, current research on this strain, including genetic sequencing, is still limited, with few studies providing comprehensive genetic analysis to link specific genetic traits with clinical outcomes in gastrointestinal and respiratory health. This gap in genetic sequencing presents a significant opportunity for future research to better understand the genetic basis of *B. longum* subsp. *infantis* B8762’s therapeutic potential.

Given the beneficial effects of B. longum subsp. infantis on gut health and immune system maturation, we hypothesize that *B. longum* subsp. *infantis* B8762 may have a significant therapeutic role in modulating both gut and respiratory microbiota in children under 2 years old. We propose that the strain may exert a therapeutic effect by reducing the severity and number of symptoms associated with gastrointestinal and respiratory illnesses. This therapeutic potential could be driven by its ability to restore microbial balance, enhance gut barrier function, and modulate immune responses, leading to the alleviation of symptoms related to dysbiosis and respiratory infections. In addition, we hypothesize that *B. longum* subsp. *infantis* B8762 may have a prophylactic effect, reducing the incidence of gastrointestinal and respiratory illnesses in young children. This preventive effect may be attributed to its ability to modulate the gut–respiratory axis, enhance immune system development, and promote a healthy microbial community, which collectively may reduce the likelihood of infections. Furthermore, we posit that the specific genetic profile of *B. longum* subsp. *infantis* B8762, particularly its ability to produce certain metabolites or interact with host immune cells, could be key to its dual therapeutic effects on both gastrointestinal and respiratory health in children.

This research seeks to investigate the therapeutic and prophylactic effects of *B. longum* subsp. *infantis* B8762 in children aged below 1 year with gastroenteritis or respiratory infections. Additionally, the study aims to explore the correlation between the genetic sequence of *B. longum* subsp. *infantis* B8762 and its clinical efficacy, identifying potential genetic markers linked to its beneficial effects. Understanding these genetic factors could enhance our ability to harness the full potential of *B. longum* subsp. *infantis* B8762 as a targeted probiotic intervention for young children, offering insights into its mechanisms of action and paving the way for more personalized probiotic therapies.

## 2. Results

### 2.1. Ethical Consideration

The study was conducted in compliance with the Declaration of Helsinki, with all procedures involving human participants approved by the IIUM & UKM Research Ethics Committee [approval numbers IIUM-IREC-2022-193 (21 December 2022) and UKM-JEP-2023-603 (27 September 2023), respectively]. It was also registered at ClinicalTrials.gov (identifier number NCT05734417).

### 2.2. Baseline Characteristic

As shown in Figure 1, 120 (71.43%) children from 168 screened potential participants were eligible to be included in the study, whereas 48 were excluded as they did not meet the inclusion criteria. Five subjects were lost to the study due to being uncontactable during the study period of 4 weeks. The final trial consisted of 57 and 58 children in the probiotic and placebo group, respectively, yielding a total of 115 children with a gender ratio (male to female) of 1.17. No adverse effects were reported throughout the study and no subjects dropped out due to any complications.

The baseline characteristics pertinent to the study outcomes showed an insignificant difference in all the sociodemographic parameters (Table 1) and anthropometry data taken at all time points (Table 2) between the probiotic and placebo subjects. This indicates a comparable population; hence, health improvement or any observed effects within the two groups are more likely attributable to the intervention rather than pre-existing differences between participants.

### 2.3. Respiratory Health

The probiotic group showed lower duration for several respiratory symptoms after 4 weeks while the placebo group did not show any changes over time (Table 3). These included fever (*p* = 0.001), cough (*p* = 0.016), nose block (*p* = 0.001), wheezing (*p* = 0.011), sore throat (*p* = 0.034), runny nose (*p* = 0.001), hoarseness (*p* = 0.005), body ache (*p* = 0.041), fatigue (*p* = 0.032), vomiting (*p* = 0.033) and nose thick mucus production (*p* = 0.008). After 4 weeks of the intervention, the probiotic group showed reduced clinical visit related to respiratory problem as compared to the placebo group (*p* = 0.009). In addition, antibiotic usage was reduced to a greater extent in the probiotic arm compared to the control group, aligning with observations from previous studies [8]. However, the specific nature of history respiratory illnesses leading to antibiotic prescriptions of all participants was not detailed in this study. The prescription of antibiotics in respiratory health is often influenced by the need to treat bacterial infections, such as bacterial pneumonia, bronchitis, or sinusitis.

### 2.4. Gastrointestinal Health

The probiotic group exhibited a shorter duration of gastrointestinal symptoms, including fever (*p* = 0.033) and fatigue (*p* = 0.038), after 4 weeks, whereas the placebo group showed no changes over the same period (Table 4). Over the 4-week period, the probiotic intervention also led to a reduction in the weekly duration of stomach aches (*p* = 0.007) and the frequency of clinical visits due to diarrhea (*p* = 0.004), with no significant changes observed in the placebo group.

### 2.5. Inflammatory Proteins Profile

Oral swab samples were evaluated for concentrations of inflammatory proteins, which included pro-inflammatory cytokines such as tumor necrosis factor-α (TNF-α), interferon-gamma (IFN-γ), and interleukin-1-beta (IL-1ß), and anti-inflammatory cytokines such as interleukin-4 (IL-4) and interleukin-10 (IL-10). Children in the placebo group showed a higher concentration of oral TNF-α after 4 weeks as compared to the probiotic group (*p* = 0.033) despite insignificant differences between groups at week 0 (Figure 2). The concentrations of oral IFN-γ and IL-10 were higher in the probiotic group as compared to the placebo group at week 0 (IFN-γ *p* = 0.042; IL-10 *p* = 0.012) but were higher in the placebo group after 4 weeks (IFN-γ *p* = 0.012; IL-10 *p* = 0.029). No changes in trends were observed between groups for concentrations of oral IL-1ß and IL-4 at baseline and at the end of study.

### 2.6. Genome Sequences of B8762

*Bifidobacterium longum* subsp. *infantis* B8762, with a chromosome size of 2.64 Mbp (Figure 3a), was isolated from the fecal sample of a healthy child in 2017 and is preserved in the China General Microbiological Culture Collection Center (CGMCC) under the accession number CGMCC No. 22765. This particular strain is recognized for its probiotic characteristics, which include resistance to the harsh conditions of the gastrointestinal tract, such as acid and bile salts, and its ability to adhere to intestinal cells. B8762 also demonstrates the capacity to inhibit the growth of pathogenic bacteria, a trait that is crucial for maintaining a healthy gut microbiome. Importantly, B8762 meets the European Food Safety Authority’s (EFSA) requirements for probiotics, as it does not exhibit antibiotic resistance, a feature that is becoming increasingly important in the face of growing antibiotic resistance in pathogenic bacteria. Furthermore, B8762 is classified as non-pathogenic and non-virulent, adding to its safety profile. It does not produce biogenic amines, which are compounds that can cause histamine-like food poisoning, and it has been shown to be non-toxic in animal cytotoxicity tests, further confirming its safety for consumption. These properties make B8762 a promising candidate for use in probiotic formulations aimed at promoting gut health and in our present study, respiratory health in infants.

Based on the genome sequence, we annotated the function of B8762 and found that its genome encompasses a wealth of functional genes associated with carbohydrate metabolisms, amino acids and their derivatives, as well as protein metabolisms (Figure 3b). Notably, among the protein metabolism-related genes, those involved in ribosome LSU bacterial (pertaining to the bacterial ribosomal large subunit [LSU]) and ribosome SSU bacterial (concerning the bacterial ribosomal small subunit [SSU]) functions are most abundant, playing pivotal roles in protein synthesis, ribosome biogenesis, and cellular growth. The LSU and SSU are essential components of the ribosome, which is the cellular machinery responsible for protein synthesis. The LSU is involved in the peptidyl transferase activity, which is the key step in the formation of peptide bonds during translation, while the SSU is involved in the initiation of protein synthesis and binds to the mRNA that carries the genetic code. The abundance of these genes in B8762 underscores its adaptability and potential for efficient protein production, which are crucial for its survival and probiotic functions within the complex environment of the human gut. This genomic feature also suggests that B8762 may have a well-developed mechanism for coping with various nutritional and environmental challenges it encounters in the gastrointestinal tract.

The analysis of carbohydrate metabolism-related genes in B8762 has revealed a multitude of diverse genes that play a role in the utilization of various carbohydrates, demonstrating significant potential in the metabolism of sucrose, the serine–glyoxylate cycle, mixed-acid fermentation, and lactic acid fermentation (Figure 4a). This metabolic versatility is crucial for the strain’s ability to adapt to different nutrient environments within the human gut. Additionally, B8762 is equipped with functional genes that enable the utilization of L-arabinose, xylose, D-ribose, D-gluconate, and ketogluconates (Figure 4b), which are important for the strain’s survival and function in diverse nutritional contexts. Furthermore, this strain contains genes related to vitamin synthesis, including the biosynthesis of vitamins B6, B1, and B2, the folate biosynthesis cluster, and biotin biosynthesis (Figure 4c). These capabilities suggest that B8762 contributes not only to gut health but also to the maintenance of overall nutritional well-being. Importantly, no virulence-related genes were annotated against the Virulence Finder 2.0 database, indicating that B8762 is non-pathogenic and safe for use as a probiotic. This comprehensive genomic profile of B8762 underscores its potential as a probiotic strain with a broad range of beneficial functions for human health.

## 3. Discussion

Our current research aimed to clarify the efficacy of a specific probiotic strain, *B. longum* subsp. *infantis* B8762, at a predetermined dosage in children. Our data provided empirical support for considering host factors, optimizing dosage, and strain specificity of probiotic interventions for children. Our study demonstrated that the administration of B8762 exerted beneficial effects on respiratory and gastrointestinal symptoms in children. Specifically, the administration of B8762 significantly reduced both the duration and frequency of symptoms associated with these infections.

Hospitalization or emergency department visits significantly impact the quality of life for children, especially those with severe symptoms, while also placing a financial and time burden on their families [13]. The key clinical manifestations that impact the quality of life for children during respiratory illness included fever, cough, nasal congestion, wheezing, sore throat, body aches, and fatigue. Meanwhile, gastrointestinal infection commonly presented with fever, vomiting/nausea, dysentery, stomach ache, poor appetite, fatigue, dizziness, headache, dehydration, and rectal pain. The decrease in symptom frequency observed after the administration of the B8762 probiotic indicated that it lowered the incidence of these infections, thus significantly reducing the number of clinical visits in children compared to the placebo group for both respiratory and gastrointestinal illnesses. These findings align with the prophylactic benefits of probiotics, as they are associated with a reduced likelihood of developing illness symptoms [14], suggesting that B8762 played a role in reducing the overall burden of disease, thereby lessening the need for medical intervention. The reduction in healthcare utilization is indicative of efficacy in mitigating the severity and recurrence of these common pediatric conditions.

The reduction in symptom duration among the probiotic group highlights the therapeutic potential of the B8762 probiotic, particularly its ability to accelerate recovery by enhancing the host’s capacity to resolve infections more rapidly. The rationale for using probiotics in infectious gut or respiratory disorders lies in their capacity to antagonize enteric pathogens through multiple mechanisms. Probiotics create an inhospitable environment for pathogens by lowering gut pH through the production of short-chain fatty acids [15] and other antimicrobial substances [16]. They also compete with pathogens for adhesion sites, preventing colonization and subsequent infection [17]. Additionally, probiotics stimulate the host’s immune system, boosting the production of immunoglobulins [18] and cytokines [19], which play critical roles in pathogen defense. Recently, a metabolomic study on *B. longum* subsp. *infantis* B8762 [20] suggested its production of compounds such as tartaric acid, procyanidin B1, oxalic acid, and ascorbic acid, which confer oxidative activity, offering protection against free radicals that can cause cell damage. These mechanisms collectively suggest that probiotics improve the overall immune resilience of the pediatric population. Future studies should focus on elucidating the precise mechanisms of action conferred by the therapeutic BB8762 strain to better understand its role in enhancing immune defense and accelerating recovery processes.

Infections and illnesses are frequently accompanied by cellular inflammation and inflammatory responses, which can be evaluated through the measurement of inflammatory proteins. The results indicated that B8762 mitigated inflammatory responses, as evidenced by the differential concentrations of pro-inflammatory and anti-inflammatory cytokines in the probiotic and placebo groups. Combining findings in a meta-analysis with 42 randomized clinical trials found a significant reduction in serum TNF-α and others inflammatory biomarkers [21]. Similarly, our data showed that children in the placebo group exhibited increased concentrations of the oral pro-inflammatory cytokine TNF-α over time. In contrast, the probiotic group did not exhibit this increase, suggesting that B8762 effectively modulated the inflammatory response associated with respiratory infections. TNF-α is a key mediator of inflammation and is often elevated during acute and chronic inflammatory conditions. Additionally, at baseline, the probiotic group had higher concentrations of the oral pro-inflammatory cytokine IFN-γ and the anti-inflammatory interleukin IL-10. Over time, these cytokines showed increased concentrations in the placebo group rather than in the probiotic group. IFN-γ is known for its role in the immune response against viral infections and its ability to activate macrophages [22], while IL-10 is a crucial anti-inflammatory cytokine that helps regulate immune responses and limit excessive inflammation [23]. The increase in pro-inflammatory cytokines TNF-α and IFN-γ in the placebo group highlighted the need for a compensatory rise in the anti-inflammatory cytokine IL-10 to buffer the inflammatory responses. This compensatory mechanism was apparent in the placebo group, which showed an increase in IL-10 over time. However, this was not observed in the B8762 group, where the levels of IL-10 remained relatively stable. This stability suggested that B8762 maintained a balanced inflammatory response, preventing the excessive inflammation often seen during respiratory illnesses in children. The ability of probiotics to modulate cytokine production and maintain a balanced inflammatory response is crucial in managing respiratory illnesses. Excessive inflammation leads to tissue damage and prolonged illnesses, whereas a balanced response promotes faster recovery and reduces the severity of symptoms. By preventing the rise in pro-inflammatory cytokines and maintaining stable levels of anti-inflammatory cytokines, probiotics offer a therapeutic advantage in managing pediatric respiratory illnesses.

Functional annotation of B8762 genome unravels its ability to metabolize a variety of host-derived and dietary carbohydrates. These include glucose derivatives like ketogluconates, the pentose sugars xylose and L-arabinose, and the glucose metabolite D-gluconate. During fermentation, these substrates produce short-chain fatty acids (SCFAs) mainly acetate, propionate and butyrate. These SCFAs, as reviewed in previous studies [24], support digestive health, boost immune function, and help protect against gastrointestinal disorders. SCFAs promote intestinal barrier integrity, exert anti-inflammatory effects by modulating cytokine production, and regulate intestinal motility while maintaining a balanced gut microbiota. Furthermore, SCFAs enhance mucin production, fortifying the intestinal protective layer as histone deacetylase inhibitors, and they influence gene expression and cellular processes while interacting with G protein-coupled receptors, affecting immune functions and hormone release. B8762, through its metabolic by-products such as SCFAs, indirectly influences children’s respiratory health via the gut–lung axis by preserving intestinal integrity, regulating immune responses, and reducing systemic inflammation [25].

Strains B8762 contains a remarkable repertoire of genes associated with vitamin synthesis, which are crucial for the health of children, particularly within the gut. The strain is equipped with genes for the biosynthesis of vitamins B6, B1, and B2, the folate biosynthesis cluster, and biotin biosynthesis. Thiamine (vitamin B1) plays a significant role in various oxidation and decarboxylation enzyme systems, particularly impacting the development of cardiovascular and nerve systems. Kunisawa et al. demonstrated that thiamine deficiency reduced the abundance of Peyer’s patches and decreased the size of B-cell follicles, leading to a reduction in naïve B-cells, thereby implicating thiamine in gut-related immuno-metabolic processes [26]. Riboflavin (vitamin B2) is involved in the metabolism of carbohydrates, fats, and amino acids, as well as cellular respiration, and is crucial for developing a healthy vision and skin. Strain B8762 aligns with other riboflavin-producing lactic acid bacteria, such as *Lactobacillus plantarum* HY7715 [27], mutant *Bifidobacterium longum* subsp. *infantis* ATCC 15,697 [28], and *Lactiplantibacillus plantarum* [29]. Vitamin B6 is a group of compounds with a defined structure, including pyridoxine (PN), pyridoxal, and pyridoxamine plays a critical role in over 100 enzyme reactions, including amino acid metabolism, neurotransmitter synthesis, and hemoglobin production [30]. It is essential for immune function and cognitive development in children, with deficiencies linked to neurological issues [31]. Vitamin B7 (biotin), a cofactor for carboxylases involved in fatty acid, glucose, and amino acid metabolism, primarily obtained through diet and, to a lesser extent, gut bacteria. Both the vitamins—B6 and B7—cannot be synthesized by the human body and rely on external sources [32,33]. Interestingly, while *Bifidobacterium* spp. lacks the biosynthetic pathway for vitamin B7, it possesses a biotin transporter, suggesting it relies on dietary or microbial sources and may compete with the host for this nutrient [34]. In contrast, certain strains, like *Bifidobacterium longum*, have been shown to produce vitamin B7 precursors, such as pimelate, in the intestine [35]. The current study demonstrates the possibility of B8762 to synthesize vitamin B7, although an extensive comparative genomics and phylogenomic analysis was required to confirm our finding. On top of that, our genomic analysis also showed that B8762 can produce and release folate extracellularly. Similarly, as observed in rat models, probiotics with folate-producing *Bifidobacterium* spp. increased plasma folate levels, confirming in vivo production and absorption [36]. Since folate is vital for DNA and RNA synthesis, particularly during growth phases, these probiotics offer a promising approach for targeted health applications. The ability of B8762 to synthesize these vitamins can be seen as an additional layer of support for the gut health and general well-being of children. These underscore the importance of vitamins produced by gut microbes in maintaining gut health, immune function, and overall metabolic balance.

B8762 also has an abundance of protein synthesis genes such as the LSU and SSU. The LSU is involved in the peptidyl transferase activity, which is the key step in the formation of peptide bonds during translation, while the SSU is involved in the initiation of protein synthesis and binds to the mRNA that carries the genetic code. High levels of these genes suggest that the strain is metabolically active, which may enhance its ability to thrive in the gastrointestinal tract [37]. This is because a higher number of ribosomal RNA genes can support increased protein synthesis, facilitating rapid growth and adaptation to varying conditions within the gut. This genomic feature also suggests that B8762 may have a well-developed mechanism for coping with various nutritional and environmental challenges it encounters in the gastrointestinal tract [38]. Efficient protein synthesis allows B8762 to thrive and exert its probiotic effects, such as competing with harmful microbes for nutrients and producing beneficial metabolites. Moreover, the ability to adapt to various nutritional and environmental challenges in the gastrointestinal tract is critical for supporting a balanced gut microbiome, which is essential for children’s overall health and development. A balanced gut microbiome contributes to the maturation of the immune system, modulation of immune responses to suppress inflammation and enhancement of the gut barrier functions. Thus, the LSU and SSU genes in B8762 are not only important for the bacteria’s own survival but also for promoting a healthy gut environment in children.

These clinical and genomic findings provided valuable insights into B8762’s broad-spectrum carbohydrate utilization, genomic stability, and probiotic properties, establishing a theoretical foundation for its potential applications in promoting both gut and respiratory health in children.

## 4. Materials and Methods

### 4.1. Trial Design, Randomization, and Blinding

This was a multicenter, randomized, double-blind, and placebo-controlled trial conducted in three university-affiliated tertiary care in Malaysia. Randomization was performed upon considering the inclusion and exclusion criteria. Qualified subjects were randomized according to 1:1 ratio to the two arms of the study according to a computer-generated list, assigned to the probiotic or placebo group with individual codes. The allocation sequence was restricted to the research statistician at the coordinating center until the completion of the study. All members of the research team (including trial and clinical staff, specimen and data analysts, as well as all participants and their parents or guardians) were unaware of the trial group assignments.

### 4.2. Intervention Product Procedure

The probiotic preparation is a lyophilized powder containing 0.5 × 10^10^ CFU of bacterial strain—*Bifidobacterium longum* subsp. *infantis* B8762 with carrier, i.e., maltodextrin, xylitol, galacto-oligosaccharides. Meanwhile, placebo contains only the carrier. All products were manufactured under an ISO9001 [39], ISO22000 [40], as well as HACCP conditions, besides, do not contain any animal-origin ingredients. Sachets of the products were packaged in such a way as to be similar in appearance of light-yellow powder, smell, and weight. Both products are kept at a storage temperature range below 30 °C according to the conditions recommended by the manufacturer.

### 4.3. Study Population

#### 4.3.1. Sample Size Calculation

The sample size was calculated for a parallel group study design involving one intervention arm and one placebo arm and was based on power design analysis. A total of 120 subjects are needed for this study, comprising of 60 total subjects in each group, including an additional 10% dropout. This calculation was based on the need for a continuous response variable from independent control and experimental subjects, with a ratio of control to subject fixed at 1:1, probability of 0.95 and Type-I error probability associated with this test of null hypothesis of 0.05. Previous data have shown that for an intervention using probiotics against reducing clinical visits in children for respiratory diseases, a standard deviation of 0.46 times within group was observed, accompanied by a reduction of 0.32 times between treatment and placebo groups [41].

#### 4.3.2. Inclusion and Exclusion Criteria

Eligible children were those aged 6- to 12-month-old, presented with respiratory illness and/or gastrointestinal problems manifesting either 2 of the following clinical symptoms: cough, fever, runny nose, sore throat, diarrhea, nausea, vomiting, bloating and abdominal pain. Subjects were recruited from the International Islamic University Malaysia and the National University of Malaysia (UKM) campuses, as well as community engagement. All participants were consented prior to the start of the study, otherwise, they will not be recruited. Other exclusion criteria included long-term medication due to certain severe illness (6 months or more) and glucose-6-phosphate dehydrogenase (G6PD) deficiency.

### 4.4. Sample Collection and Data Analyses

#### 4.4.1. Demographic Data and Questionnaires

Eligible subjects who met all the inclusion and exclusion criteria were given three types of questionnaires; (I) demographical questionnaire (collected at baseline week 0), (II) respiratory health questionnaire (collected at baseline week 0, and week 4), and (III) gut health questionnaire (collected at baseline week 0, and week 4). All questionnaires were validated and translated to the Malay language [42].

#### 4.4.2. Enzyme-Linked Immunosorbent Assay (ELISA) for Inflammatory Proteins

Oral swab samples were collected from each child at week 0 and week 4, using swabs from the inside of both left and right cheeks, one immediately after the other. These samples were collected by the medical team, placed in sterile saline, and frozen at −80 °C until further analysis. To determine the concentrations of pro- and anti-inflammatory cytokines (TNF-α, IFN-γ, IL-1β, IL-4, and IL-10), enzyme-linked immunosorbent assay (ELISA) kits (Sunlong Biotech, Hangzhou, China) were used, following the manufacturer’s instructions. For each cytokine, a double antibody sandwich ELISA technique was applied, where specific antibodies pre-coated on high-affinity ELISA plates captured the target analytes. After incubation with biotinylated detection antibodies and HRP-conjugated secondary antibodies, unbound substances were washed away, and a TMB colorimetric substrate was added. The plates were incubated in the dark for color development, and the reaction was terminated using a stop solution. Absorbance was measured at 450 nm with a reference wavelength of 570–630 nm. The cytokine concentrations were determined by comparing the optical density values of the samples to the standard curves.

#### 4.4.3. B8762 Genome Sequencing and Assemblies

The genome sequencing of B8762 was performed using the Illumina NovaSeq6000 platform (Illumina, Inc., San Diego, CA, USA), generating 150 bp paired end reads of approximately 1 Gbp of data. The raw sequencing reads were processed and filtered using fastp software (v0.23.0, Peking University, Beijing, China), followed by assembly with spades (v3.15.2, St. Petersburg genome center, Saint Petersburg, Russia). Glimmer (v0.92.3, Department of Computer Science, Johns Hopkins University, Baltimore, MD, USA) was employed for CDS prediction, tRNA-scan-SE (v2.0.12, Department of Biological Sciences, Stanford University, Stanford, CA, USA) for tRNA prediction, and Barrnap (v0.9, Department of Computer Science, University of Edinburgh, Edinburgh, Scotland, UK) was used for rRNA prediction. The predicted CDSs were annotated using sequence alignment tools and databases, including NR, Swiss-Prot, Pfam, GO, COG, and KEGG. Tools such as BLASTP (National Center for Biotechnology Information (NCBI), Bethesda, MD, USA), DIAMOND (v0.9.24, Department of Computer Science, University of Maryland, College Park, MD, USA), and HMMER (v3.4, Howard Hughes Medical Institute, Department of Biostatistics, University of California, Berkeley, CA, USA) for sequence alignment and functional annotation.

#### 4.4.4. Statistical Approaches

Intention-to-treat analysis was performed using SPSS software version 20.0 (SPSS Inc., Chicago, IL, USA). Considering the skewed distribution and non-parametric nature of our data, differences between groups were compared using the Mann–Whitney U test, while nominal data were compared using the Chi-squared test. All tests were two-sided with *p* < 0.05 as considered statistically significant and data are presented as mean values ± standard error unless stated otherwise.

## 5. Conclusions

The administration of the *Bifidobacterium longum* subsp. *infantis* B8762 probiotic strain demonstrates significant therapeutic and prophylactic benefits in managing respiratory and gastrointestinal illnesses in young children. Our study demonstrates the therapeutic potential of *B. longum* subsp. *infantis* B8762 in managing respiratory and gastrointestinal symptoms in children, with significant reductions in the incidence, duration, and severity of infections, as well as decreased reliance on medical interventions such as hospitalizations. The observed therapeutic effects of B8762 can be attributed to its immune-modulating properties, including its ability to balance inflammatory responses. Specifically, B8762 effectively regulated pro-inflammatory cytokines such as TNF-α and IFN-γ, maintaining stable levels of anti-inflammatory cytokines like IL-10. This balanced immune response is crucial for managing respiratory infections, as it reduces the risk of excessive inflammation, which can lead to prolonged symptoms and tissue damage. Furthermore, the fermentation of carbohydrates by B8762 produces short-chain fatty acids (SCFAs) that promote intestinal health, enhance immune function, and help protect against gastrointestinal disorders. The probiotic’s genomic capability to synthesize essential vitamins like thiamine further supports immune modulation, suggesting its role in maintaining overall metabolic balance. B8762, through their anti-inflammatory effects and interactions with immune pathways, further support the gut–lung axis, indirectly benefiting respiratory health in children. These findings establish B8762 as a promising intervention for pediatric health, with potential economic benefits by reducing healthcare utilization and improving quality of life. Future research should focus on further elucidating its mechanisms of action, optimal dosage, and potential for synergistic therapies in managing common childhood infections. Therefore, B8762’s ability to promote intestinal integrity, modulate immune responses, and regulate inflammation suggests its strong potential as a therapeutic tool in both respiratory and gastrointestinal disorders. By expanding our understanding of its mechanisms, B8762 could play a key role in advancing pediatric health, reducing the burden of infectious diseases and improving quality of life for children.

## 6. Limitation and Future Study

This study has several limitations, including its short 4-week duration, which may not capture long-term effects, and the potential influence of uncontrolled confounding factors such as diet and environmental exposures. Reliance on saliva for cytokine measurements, while convenient, may not fully reflect systemic immune responses due to sensitivity issues, time-sensitive fluctuations, and interference from oral microbiota. Additionally, the use of parent-reported questionnaires introduces potential bias and subjective inaccuracies. Future research should address these limitations by conducting longer-term studies with diverse populations, integrating more robust biomarkers, and controlling confounding factors to validate findings. Despite these limitations, the study is novel in demonstrating the therapeutic potential of *B. longum* subsp. *infantis* B8762 in improving respiratory and gastrointestinal health in children by modulating immune responses, promoting intestinal health, and supporting the gut–lung axis, offering a promising approach to pediatric care.

## Figures and Tables

**Figure 1 ijms-26-01323-f001:**
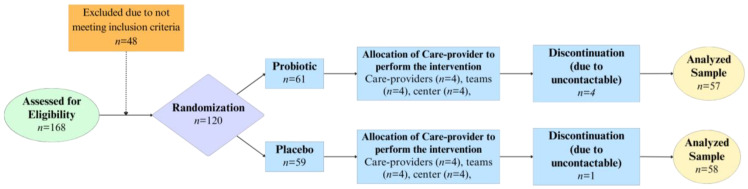
Study consort flowchart detailing the process of subject recruitment, randomization, and allocation of care provider, before proceeding with patient follow-up and sample or data analysis.

**Figure 2 ijms-26-01323-f002:**
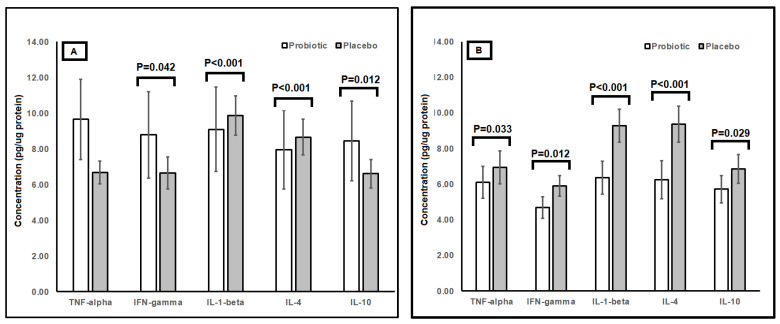
Concentrations of inflammatory proteins as obtained from oral swab samples of children (*n* = 115) randomly assigned to a double-blind administration of 4 weeks with either placebo (*n* = 58; grey) or probiotic *B. longum* subsp. *infantis* B8762 (*n* = 57; white). Results are expressed as means; error bars (SEM). (**A**) Comparison of inflammatory protein between probiotic and placebo taken at week 0. (**B**) Comparison of inflammatory protein between probiotic and placebo taken at week 4.

**Figure 3 ijms-26-01323-f003:**
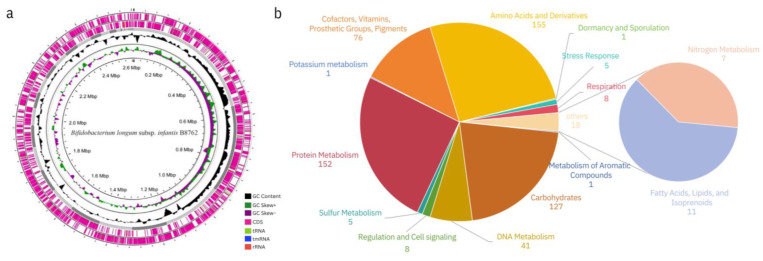
(**a**) Circular map of whole genome of *Bifidobacterium longum* subsp. *infantis* B8762. From the innermost to the outermost circle, the first circle represents the genome size. The second and third circles illustrate the GC skew values, calculated using the formula G − C/G + C, which help identify the leading and lagging strands; typically, the leading strand has a GC skew > 0, and the lagging strand has a GC skew < 0. These circles also assist in pinpointing the replication origin (minimum cumulative offset) and the replication terminus (maximum cumulative offset), particularly important for circular genomes. The fourth circle, in black, represents the GC content: regions with higher GC content than the genome average is shown outward, while regions with lower GC content are shown inward, with higher peaks indicating greater deviations from the average GC content. The outermost two layers display the CDSs (coding sequences) on the positive and negative strands, with different colors representing distinct COG functional categories. (**b**) Functional annotation information of B8762. Functional classification of B8762 genome representing the distribution of genes based on their annotations to terms in RAST.

**Figure 4 ijms-26-01323-f004:**
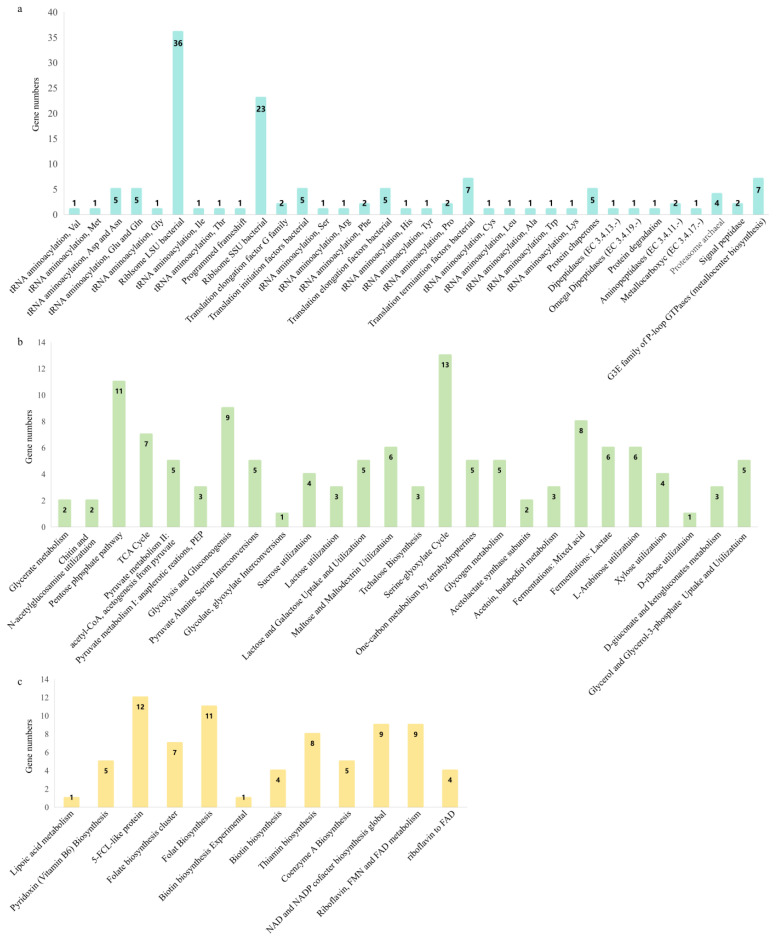
The gene numbers of protein metabolism (**a**), carbohydrate metabolism (**b**), and microbial metabolism (**c**) in *Bifidobacterium longum* subsp. *infantis* B8762.

**Table 1 ijms-26-01323-t001:** Baseline characteristics of children, *n* = 115, randomly assigned to a double-blind administration with either placebo (*n* = 58) or *B. longum* subsp. *infantis* B8762 probiotic (*n* = 57).

Baseline Characteristics	Placebo	Probiotic	*p*-Value *
Gender, % (*n*)			
Male	53 (31)	54 (31)	0.920 **
Female	47 (27)	46 (26)	
Age (months)	2.78 ± 0.51	3.51 ± 0.48	0.106
Body weight (kg)	8.17 ± 0.34	8.22 ± 0.26	0.663
Height (cm)	70.67 ± 1.79	71.74 ± 1.27	0.549
Body mass index (kg/cm^2^)	17.10 ± 1.11	16.03 ± 0.42	0.799
Smokers in family	0.47 ± 0.07	0.42 ± 0.07	0.633
History of food allergy	0.12 ± 0.04	0.12 ± 0.04	0.972
Hospitalization for the past 12 months	0.19 ± 0.05	0.23 ± 0.06	0.614
Defecation frequency (per week)	14.14 ± 1.21	11.93 ± 0.87	0.284
Antibiotic intake	0.05 ± 0.03	0.04 ± 0.02	0.663
Having pets at home	0.24 ± 0.06	0.40 ± 0.07	0.064
	% (*n*)	% (*n*)	*p*-Value **
Family Income Status			
Low (<RM 2900)	43.10 (25)	29.82 (17)	0.113
Medium (RM 3000–RM 9000)	53.45 (31)	57.89 (33)	
High (>RM 10,000)	3.45 (2)	12.28 (7)	
House location			
Urban	58.62 (34)	59.65 (34)	0.911
Suburban	41.38 (24)	40.35 (23)	
Type of residence			
Terrace/Story/Bungalow	81.03 (47)	73.68 (42)	0.346
Apartment	18.97 (11)	26.32 (15)	

* *p*-value obtained via Mann–Whitney U-test; ** *p*-value obtained via Chi-squared test.

**Table 2 ijms-26-01323-t002:** Anthropometry data of children, *n* = 115, taken at week 0 and week 4 randomly assigned to a double-blind administration with either placebo (*n* = 58) or *B. longum* subsp. *infantis* B8762 probiotic (*n* = 57).

Parameters	Placebo			Probiotic		
	Week 0	Week 4	*p*-Value *	Week 0	Week 4	*p*-Value *
Weight (kg)	8.16 ± 0.34	8.42 ± 0.33	0.409	8.22 ± 0.26	8.68 ± 0.25	0.232
Height (cm)	70.67 ± 1.79	71.56 ± 1.69	0.705	71.74 ± 1.26	73.94 ± 1.40	0.256
BMI	17.10 ± 1.10	16.70 ± 0.66	0.643	16.02 ± 0.41	16.17 ± 0.47	0.919

All data are presented as mean values ± standard error; * *p* < 0.05.

**Table 3 ijms-26-01323-t003:** Subjects’ responses for respiratory incidence questionnaire, *n* = 115, taken at week 0 and week 4 randomly assigned to a double-blind administration with either placebo (*n* = 58) or *B. longum* subsp. *infantis* B8762 probiotic (*n* = 57).

Parameters	Placebo			Probiotic		
	Week 0	Week 4	*p*-Value	Week 0	Week 4	*p*-Value
# Clinical Visit for Current Respiratory Problem:						
Yes (*n*, %)	32, 55%	29, 50%	0.577	25, 44%	12, 21%	0.009 *
No (*n*, %)	26, 45%	29, 50%		32, 56%	45, 79%	
# Antibiotic Use for Current Respiratory Problem:						
Yes (*n*, %)	9, 16%	7, 12%	0.590	9, 16%	5, 9%	0.254
No (*n*, %)	49, 84%	51, 88%		48, 84%	52, 91%	
Number of Days with Respiratory Symptoms (Per Week) In the Past Month:						
Fever	3.09 ± 0.39	3.01 ± 0.40	0.839	1.66 ± 0.26	0.67 ± 0.19	0.001 *
Cough	3.71 ± 0.39	3.49 ± 0.40	0.660	2.50 ± 0.31	1.52 ± 0.26	0.016 *
Sneezing	3.18 ± 0.38	3.03 ± 0.39	0.639	1.50 ± 0.25	1.00 ± 0.17	0.392
Nose block	3.74 ± 0.41	3.47 ± 0.41	0.661	2.09 ± 0.32	0.91 ± 0.26	0.001 *
Wheezing	2.77 ± 0.41	2.59 ± 0.41	0.664	0.98 ± 0.23	0.45 ± 0.19	0.011 *
Sore throat	2.69 ± 0.41	2.47 ± 0.41	0.639	0.73 ± 0.22	0.35 ± 0.18	0.034 *
Runny nose	4.20 ± 0.39	4.02 ± 0.39	0.683	3.34 ± 0.37	1.84 ± 0.35	0.001 *
Poor appetite	3.01 ± 0.39	2.91 ± 0.40	0.810	1.53 ± 0.29	0.86 ± 0.22	0.069
Hoarseness	2.48 ± 0.40	2.43 ± 0.40	0.812	1.13 ± 0.25	0.54 ± 0.21	0.005 *
Body ache	2.36 ± 0.40	2.38 ± 0.41	0.930	0.50 ± 0.19	0.18 ± 0.13	0.041 *
Fatigue	2.77 ± 0.41	2.81 ± 0.41	0.965	0.85 ± 0.23	0.45 ± 10.19	0.032 *
Vomiting	2.41 ± 0.40	2.39 ± 0.40	0.903	0.78 ± 0.20	0.43 ± 0.18	0.033 *
Headache	2.34 ± 0.41	2.28 ± 0.41	0.802	0.43 ± 0.18	0.18 ± 0.13	0.071
Thick mucus	2.73 ± 0.41	2.56 ± 0.41	0.670	1.20 ± 0.25	0.63 ± 0.23	0.008 *
Pain swallowing	2.60 ± 0.41	2.58 ± 0.42	0.907	0.38 ± 0.15	0.30 ± 0.17	0.161
# Past Month Respiratory Problem:						
Yes (*n*, %)	23, 40%	24, 41%	0.850	13, 23%	6, 11%	0.079
No (*n*, %)	35, 60%	34, 59%		44, 77%	51, 89%	

All data are presented as mean values ± standard error; * *p* < 0.05; # *p*-value obtained via Chi-squared test.

**Table 4 ijms-26-01323-t004:** Subjects’ responses for gastrointestinal incidence questionnaire, *n* = 115, taken at week 0 and week 4 randomly assigned to a double-blind administration with either placebo (*n* = 58) or *B. longum* subsp. *infantis* B8762 probiotic (*n* = 57).

Parameters	Placebo			Probiotic		
	Week 0	Week 4	*p*-Value	Week 0	Week 4	*p*-Value
Period (days) for stomach discomfort (such as wind and bloating) per week	0.95 ± 0.14	0.81 ± 0.12	0.576	1.19 ± 0.19	0.84 ± 0.12	0.442
Period (days) for stomach ache per week	3.79 ± 1.75	0.76 ± 0.16	0.763	2.05 ± 1.05	0.54 ± 0.18	0.007 *
Defecation times per week	12.10 ± 1.04	11.14 ± 0.91	0.636	11.32 ± 0.95	10.79 ± 0.87	0.591
# Occurrence of Diarrhea in The Past Month:						
Yes (*n*, %)	28, 48%	30, 52%	0.710	21, 37%	13, 23%	0.101
No (*n*, %)	30, 52%	28, 48%		36, 63%	44, 77%	
Past month diarrhea incident (number of times per child)	2.03 ± 0.55	2.10 ± 0.55	0.858	0.53 ± 0.11	0.68 ± 0.27	0.160
Past month clinical visit due to diarrhea incident (number of times per child)	0.16 ± 0.05	0.14 ± 0.05	0.794	0.23 ± 0.06	0.04 ± 0.02	0.004 *
Number of Days with Gastrointestinal Symptoms (Per Week) In the Past Month:						
Fever	0.09 ± 0.06	0.03 ± 0.03	0.554	0.45 ± 0.16	0.16 ± 0.12	0.033 *
Vomit	0.02 ± 0.02	0.02 ± 0.02	1.000	0.18 ± 0.06	0.09 ± 0.05	0.225
Dysentery	0.62 ± 0.26	0.62 ± 0.26	1.000	0.12 ± 0.04	0.04 ± 0.02	0.084
Stomach ache	0.72 ± 0.17	0.68 ± 0.17	0.956	0.19 ± 0.06	0.12 ± 0.07	0.106
Nausea	0.00 ± 0.00	0.00 ± 0.00	1.000	0.11 ± 0.04	0.02 ± 0.02	0.052
Poor appetite	0.34 ± 0.12	0.36 ± 0.12	0.824	0.57 ± 0.20	0.44 ± 0.19	0.251
Fatigue	0.41 ± 0.17	0.42 ± 0.17	0.789	0.36 ± 0.12	0.11 ± 0.08	0.038 *
Dizziness	0.39 ± 0.17	0.39 ± 0.17	1.000	0.11 ± 0.04	0.02 ± 0.02	0.052
Headache	0.39 ± 0.17	0.39 ± 0.17	1.000	0.11 ± 0.04	0.02 ± 0.02	0.052
Dehydration	0.41 ± 0.17	0.41 ± 0.17	1.000	0.12 ± 0.04	0.05 ± 0.03	0.187
Rectal pain	0.39 ± 0.17	0.41 ± 0.17	0.773	0.11 ± 0.04	0.02 ± 0.02	0.052
Past month diarrhea incident (number of days)	0.97 ± 0.18	0.98 ± 0.17	0.787	0.91 ± 0.19	0.49±	0.081
Past month diarrhea incident (frequency per day)	1.28 ± 0.21	1.41 ± 0.21	0.664	1.25 ± 0.23	0.68±	0.070

All data are presented as mean values ± standard error. * *p* < 0.05; # *p*-value obtained via Chi-squared test.

## Data Availability

The original contributions presented in this study are included in the article only. Further inquiries can be directed to the corresponding author(s).

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
