# Peer review of "Therapeutic Potential of Bifidobacterium longum subsp. infantis B8762 on Gut and Respiratory Health in Infant"

_ijms, 2025, doi:10.3390/ijms26031323_

Round 1

Reviewer 1 Report

Comments and Suggestions for Authors

Journal: IJMS

Full Title: Therapeutic Potential of Bifidobacterium longum subsp. Infantis B8762 on Gut and Respiratory Health in Infant

Article type: Article

General comments

Thank you for asking me to review this article aimed to assess the efficacy of the probiotic Bifidobacterium longum subsp. Infantis B8762 in reducing the duration and frequency of respiratory and gastrointestinal infections in young children. The issue is relevant and the study interesting. Every section is well-structured. English is fluent. However, I have some suggestions to the Authors. Herein, I reported point-to-point comments to the manuscript.

Minor comments

Abstract: the abstract is concise and well-written.

Introduction: This section discusses epidemiological data about respiratory and gastrointestinal infections in pediatric population. Moreover, it provides information on the increasing relevance of probiotic products in managing these conditions.

- This section is well-written and the purpose of the study is clear. However, the increasing importance regarding the use of probiotic products in infectious diseases in children is related to the identification of microbiota changes in gut and lungs. The role of microbiota in the immune regulation in lungs is a recent finding and Authors should emphasize its role. I suggest referring to the following manuscript: “Gambadauro A, Galletta F, Li Pomi A, Manti S, Piedimonte G. Immune Response to Respiratory Viral Infections. Int J Mol Sci. 2024;25(11):6178. Published 2024 Jun 4. doi:10.3390/ijms25116178”.

Methods:

- This section is reported at the end of the manuscript. However, I suggest including it between introduction and results, to increase the comprehension of the results.

- Line 369: “Any members of the research team including; trial and clinical staff…” needs to be modified in “Any members of the research team (including trial and clinical staff, […]) were unaware of the trial-group assignments”.

- Line 401: Why did the Authors decide to exclude specific patients with G6PD deficiency?

- Authors should add an “ethics” section.

Results This section is clear and figures and tables are well-reported.

Discussion: This section is interesting and well-structured.

- Recent research has shown the importance of Streptococcus salivarius in upper respiratory tract infections. However, Authors did not discuss its role. I suggest comparing their results with those reported in the following manuscript: “Manti S, Parisi GF, Papale M, et al. Bacteriotherapy with Streptococcus salivarius 24SMB and Streptococcus oralis 89a nasal spray for treatment of upper respiratory tract infections in children: a pilot study on short-term efficacy. Ital J Pediatr. 2020;46(1):42. Published 2020 Apr 3. doi:10.1186/s13052-020-0798-4

- Authors did not provide a section about the limitations of their study. I suggest adding this section due to its utility for future studies.

Reviewer 2 Report

Comments and Suggestions for Authors

·       Thank you for your efforts

1-The manuscript needs radical organization as methods have to come before the results and conclusions to follow the discussion

2- The problem is not fully explained  as well as the  relation to genetic sequences

3- The study lacks the hypothesis that research needs to prove

4- The objective is not fully explained and does not use precision terminology : [ The objective of the present study was to investigate the therapeutic and prophylactic effect of these probiotic strains in managing respiratory and gastrointestinal illnesses in young children].

·        age < 1 year is infant

·        Methods and results did not show what was therapeutics and what was the prophylactic effect.

·        The objective did not show any clue about the genetic sequences

·        what is meant by these strain

Authors need to adjust all of these points

5. Methods:

·        It is better to explain what you are going to do at the beginning, add the duration of the study, and ethical approval.  Also, item 4.4.4. B8762 Genome Sequencing and Assemblies should come before the Statistical Approaches.

·        how accurate are the Oral swab samples in measuring acute infection? can you elaborate as usually better for chronic infection

·        How you avoid bias from oral infection /inflammation

·         Each studied inflammatory cytokine ( TNF, IFN gamma IL-1 beta, IL4, iL-10, has a different kit, please indicate in the methods the source for each one.

·        Mention in the methods line 414, that you studied immunoglobulin, can you add to the results?

 6-Results

 ·        It is not clear how you interpret some symptoms in infants less than one-year-old ( body aches – headaches –pain in swallowing –nose block –stomach aches –rectal pain – and dizziness ), and how infants express these symptoms.  

·        Table 1: data expressed as mean and SD / however the used test was Mann Whitney and not a T-test. ( A part of the use of antibiotics results looks normally distributed ) please explain

·        What was the nature of respiratory problems, some need the use of antibiotics, so explain in the result section

 ·        Figure 2: I suggest a table to show the exact values of the studied inflammatory protein /Mean /SD or median and use a significant test. It will be better to visualize the data

 ·        Resize Fig 3 and 4 especially 4

 7-Discussion and conclusion :

·        It is appropriate, but

·        What was the limitation of this study?

·        What the study added ?

8-References

·        To review with j style

Round 2

Reviewer 2 Report

Comments and Suggestions for Authors

Thank you for your efforts and clear adequate responses  to all my comments .